# The longitudinal relations between physical activity, inflammation, and depression in the Health and Retirement Study

Andrew Levihn-Coon *, Jasper A.J. Smits, Christopher G. Beevers

Department of Psychology and Institute for Mental Health Research, University of Texas at Austin, Austin, Texas, United States of America

* alevihncoon@utexas.edu

## Abstract

Physical activity appears associated with lower depression in treatment and epidemiological studies but specific biological mechanisms remain unclear. Research supports inflammation being positively associated with depression and negatively associated with physical activity, suggesting it could mediate physical activity's effect on depression. This study examined longitudinal associations between physical activity, inflammation, and depression symptoms in 13,461 older adults (59% women, mean age = 68) using the Health and Retirement Study dataset. Depression, physical activity, and inflammation (high sensitivity C-reactive protein (hsCRP)) were measured three times, four years apart. We used random intercept cross-lagged panel models (RI-CLPM) to test for stable between-person and within-person associations over time. Between-person results found higher physical activity was associated with lower hsCRP (β = −0.40, SE = 0.087, p < 0.001) and depression scores (β = −0.48, SE = 0.137, p < 0.001), suggesting physical activity is associated with lower inflammation and depressive symptoms. Conversely, higher hsCRP was correlated with higher depression scores (β = 0.19, SE = 0.20, p < 0.001). Within-person results revealed no support for change in inflammation as a longitudinal mediator of the relation between physical activity and change in depressive symptoms. Findings suggest robust relations among trait levels of physical activity, inflammation, and depression, but little support for change in inflammation as a longitudinal mediator of the physical activity and depression association. Future studies with additional assessments and shorter intervals may clarify the temporal associations among physical activity, inflammation, and depression.

## Introduction

Physical activity is increasingly supported by research as a viable alternative intervention for both the prevention and treatment of depression [1–6]. Meta-analyses of correlational studies have shown that physical activity is associated with improved mental health and reduced depression [7,8]. Additionally, meta-analyses of randomized controlled trials (RCTS) have found that physical activity can lead to significant improvements in depressive symptoms, demonstrating effects ranging from medium to large, when compared to either control

**Data availability statement:** The data used in this study are from the Health and Retirement Study (HRS), which is publicly available and conducted by the University of Michigan (https://hrs.isr.umich.edu/data-products). The HRS data conditions of use (https://hrsdata.isr.umich.edu/data-products/conditions-of-use?_gl=1*tutk3d*_ga*NTczNjI2OTIuMTcxMTM4NjY3OQ..*_ga_FF28MW3MW2*MTcyNDI3Nzc2OC42LjEuMTcyNDI3Nzk3Mi4wLjAuMA) prohibit redistribution of any HRS data product to any third party. To obtain the HRS Public Release data files one must register as a user on their website and then one may download the files. In order to obtain access to the HRS Sensitive Health Data that contains the biomarker data used in this manuscript's analysis, one must additionally submit a Sensitive Health Order Form (https://hrsdata.isr.umich.edu/data-products/sensitive-health/order-form) to request access to the data through HRS. The code utilized to conduct the analysis found in the manuscript can be found on our laboratory's dataverse: https://dataverse.tdl.org/dataverse/hrs-pa-inflammation-depression. The HRS data files used to conduct our analysis were the "RAND HRS Longitudinal File 2018 V2 Public Use Dataset" as well as "Biomarker Data" files from 2006-2016.

**Funding:** The authors received no specific funding for this work.

**Competing interests:** Dr. Smits has received grants from the National Institutes of Health, the Department of Defense, and the Trauma Research and Combat Casualty Care Collaborative Prevention. He has received personal fees from Big Health and various universities for consulting, and from Elsevier and the American Psychological Association for editorial activities. Dr. Smits also has equity interest in Earkick, and has received royalty payments from various publishers. The terms of these arrangements have been reviewed and approved by the University of Texas at Austin in accordance with its conflicts of interest policies. Christopher Beevers has received funding for his research from the National Institutes of Health, Brain and Behavior Foundation, Aiberry Inc., and other not-for-profit entities. He has received income from the Association for Psychological Science for his editorial work and from Orexo, Inc. for serving on a Scientific Advisory Board related to digital therapeutics. Dr. Beevers' financial disclosures have been reviewed and approved by the University of

[5,9–11] or no treatment [5,9,12]. Meta-analyses further confirm that these benefits are comparable to those of standard psychotherapies [5,6,9,10,13] and pharmaceutical treatments [5,6,10], especially in cases of mild to moderate depression. The ideal dose and timing metrics of physical activity prescriptions for depression are still the focus of current research, but there is growing support that physical activity has a larger antidepressant effect when it is administered at a higher intensity [5,6,14–16].

Though the positive impact of physical activity on depression is well documented, the underlying biological mechanisms by which it alleviates depressive symptoms are not fully understood [3,17]. Physical activity can influence various biological mechanisms that are also involved in the development of depression, including neuroplasticity [18], oxidative stress [19], the endocrine system [20], and inflammation [3]. While each of these biological mechanisms may play a significant role in the nexus between physical activity and depression, this paper focuses on inflammation, and more specifically, the inflammatory marker, C-reactive protein (CRP). CRP is a pentameric acute-phase reactant protein primarily produced by hepatocytes in the liver as part of the body's acute-phase response to inflammation [21]. CRP levels can be easily measured via blood testing, including its high-sensitivity variation (hsCRP) [22]. Measuring CRP levels is commonly utilized in medical settings to indicate the presence of infection, ongoing disease processes, and persistent low-level inflammation [21,23].

For over three decades research has indicated that inflammation could be implicated in the underlying mechanisms of depression [24]. Systemic immune activation is observed in cases of major depression, as evidenced by elevated levels of pro-inflammatory cytokines [25–29] and alterations in the acute phase protein response, specifically through the increase of positive acute phase proteins (such as CRP) and the reduction of negative acute phase proteins [30–33]. Additionally, evidence indicates that systemic inflammation could be a risk factor for depression. In animal studies, the systemic introduction of pyrogens into mice triggers sickness behavior that closely resembles depressive symptoms observed in humans such as fatigue, decreased motivation, decreased appetite and weight, anhedonia, memory deficits, sleep disturbance, and impairments in cognitive and social functioning [34–37]. Additionally, autoimmune diseases and infections in early life are associated with an increased likelihood of developing depression later in adulthood [38] and depression frequently co-occurs with conditions characterized by elevated levels of inflammation, including obesity, cardiovascular diseases, certain nutritional deficiencies, and smoking [38–40].

Within the literature, there is further support for the link between the specific inflammatory biomarker, CRP, and depression. Comprehensive meta-analyses reveal that depressive individuals often show increased levels of various inflammatory markers, including CRP [41,42]. Indeed, when specifically examining the link between CRP and depression, a recent meta-analysis that examined 56 studies found that in most studies a positive association existed between elevated CRP levels and depression and that CRP levels tend to be associated with depressive severity [43]. Additionally, elevated levels of hsCRP have been associated with a higher risk of developing depression, suggesting that hsCRP may serve as a prognostic marker for major depressive disorder (MDD) [44]. The meta-analysis also found that consistently high levels of CRP correlate with a greater likelihood of experiencing symptoms of depression in later life [45]. Furthermore, meta-analyses suggest that medications with inflammatory effects can potentially contribute to the development of depressive symptoms [46], while those with anti-inflammatory properties may reduce the symptoms of depression [47]. While there is convincing research linking depression and inflammation, a definitive causal relation with specific biomarkers, such as CRP, has yet to be conclusively identified [48].

A strong relation between inflammation and physical activity is also supported by recent research. Multiple meta-analyses have reported that engaging in physical activity

Texas at Austin in accordance with its conflict-of-interest policies. There are no patents, products in development or marketed products to declare. This does not alter our adherence to PLOS Mental Health policies on sharing data and materials.

interventions can lead to reductions in several biomarkers of inflammation in the bloodstream, including CRP [49–52]. Additionally, several prospective cohort studies have shown a correlation between low levels of physical activity and higher levels of inflammation markers such as CRP [53,54]. However, research specifically examining the relation between CRP and physical activity is much more mixed. A meta-analysis by Michigan et al. [52] found that significant reductions in CRP levels were found in 11 of 25 aerobic-based trials, neither of two resistance-based trials, and two of five combination trials. Of note, however, of the 32 studies included in the analysis, 15 reported sample sizes below N = 50, suggesting that the studies likely possessed insufficient statistical power to detect relevant effects. Research on the relation between physical activity and CRP in elderly populations also reported mixed findings. Three clinical trials have found reductions in CRP levels following implemented physical activity interventions in elderly populations [55–57], and one meta-analysis of resistance training in elderly populations found that resistance training was effective in reducing CRP levels [58]. However, another found no significant CRP reductions after the implementation of an aerobic-based regimen, though the authors noted that the absence of reductions in CRP levels could be attributed to insufficient study duration and minimal loss of body fat, which might have impacted the expected outcomes [59]. That said, a more recent meta-analysis by Fedewa et al. [50] of 83 randomized and non-randomized controlled trials found that physical activity was associated with a decrease in CRP levels, irrespective of age or sex of the participant, but that greater reductions in CRP levels occurred with a decrease in BMI or % body fat.

Growing evidence suggests that physical activity can mitigate inflammation, which in turn, plays a crucial role in both the development and management of depression [16,60,61]. Yet, there remains a significant gap in direct studies examining the anti-inflammatory effects of physical activity among individuals suffering from depression [3]. The limited studies that do exist have reported mixed findings. A recent meta-analysis by Schuch et al. [11] identified three studies [62–64] that were focused on the long-term impact of physical activity on inflammation in depressed samples and each concluded that physical activity did not result in notable alterations in inflammatory markers among individuals with depression. However, one recent RCT found increases in anti-inflammatory markers in the physical activity conditions relative to other groups as well as reductions in depression symptoms, though CRP was not reduced outside of a subgroup analysis of patients with potentially higher cardiovascular risk [65]. Another RCT indicated that engaging in physical activity led to decreases in both the pro-inflammatory marker IL-6 and depressive symptoms in individuals diagnosed with depression [66], however, CRP levels were not available in this analysis. Lastly, in a recent large cross-sectional study of 18,453 adults using NHANES data to investigate the relations between physical activity, depression, and inflammation, Guo and Le [16] found that higher physical activity levels were associated with lower depression risk and reduced levels of inflammatory markers, including neutrophil count, white blood cell count (WBC), neutrophil-to-lymphocyte ratio (NLR), and the systemic immune-inflammation index (SII), with inflammation partially mediating the relationship between physical activity and depression. Considering these mixed findings, more research is needed to investigate if physical activity reduces inflammation in adults and if such inflammatory changes reduce depressive symptoms.

Moreover, older adults are particularly vulnerable to both depression and chronic low-grade inflammation, making them a critical population for investigating the interplay among physical activity, inflammation, and mental health. A meta-analysis encompassing 42 studies with 57,486 participants reported a pooled prevalence of depression among older adults of 31.74%, with factors such as physical health and social support playing significant roles [67]. Chronic low-grade inflammation, often called "inflammaging," is also prevalent

in this population and has been linked to age-related diseases, including cardiovascular disease, diabetes, neurodegenerative disorders, and depression [68]. Physical inactivity has been shown to exacerbate chronic inflammation in older adults, elevating levels of pro-inflammatory markers such as CRP and IL-6 [69]. Conversely, regular physical activity mitigates these issues by reducing inflammatory markers. Recent systematic reviews have found that both resistance and aerobic training effectively modulate inflammatory markers [70], and that combining physical activity with dietary supplementation further improves inflammatory profiles in older adults [71]. Investigating these relationships in older adults is subsequently crucial for developing interventions to enhance both physical and mental health amongst this population.

In the current study, we utilized data from the Health and Retirement Study (HRS), a longitudinal prospective study of over 22,000 American adults over 50 years old, to examine the longitudinal relations between physical activity, the inflammatory marker hsCRP, and depression measured at three time points separated by four years each. We utilized random intercept cross-lagged panel models (RI-CLPM), a fully dynamic structural equation model [72], to discern the directional influences between physical activity, hsCRP levels, and depressive symptoms over time.

Importantly, RI-CLPM disaggregates within- and between-person variance, which allows for examining auto-regressive and cross-lagged effects at the within-person level while simultaneously controlling for the stable, trait-like associations among constructs at the between-subjects level [73]. This distinction is crucial in our study because individuals inherently differ in baseline levels of physical activity, inflammation, and depressive symptoms (e.g., genetic predispositions, personality, overall health). By modeling random intercepts for each variable, RI-CLPM ensures that the cross-lagged paths reflect only within-person fluctuations over time—thereby helping us more confidently infer potential causal pathways among physical activity, inflammation, and depression. Not separating the within- and between-person variance, as is done in traditional analytic approaches such as the Cross-Lagged Panel Model (CLPM), can lead to inflated or inaccurate estimates of the cross-lagged relationships [73]. This has not previously been done, suggesting that the within-person, longitudinal relationships between physical activity, hsCRP, and depression symptoms may not have been precisely estimated in the prior literature. Evaluating both the between- and within-person effects is critical for understanding the underlying mechanisms that drive associations between physical activity, inflammation, and depression. For example, while between-person associations may highlight general population-level trends, within-person analyses provide unique insights into how changes in an individual's behavior or physiological state over time influence other outcomes. These insights are particularly valuable for tailoring interventions and understanding causal processes in a naturalistic context. By employing RI-CLPM, this study aims to advance the literature by providing a more nuanced understanding of these relationships.

The primary aim of the current study was to investigate if inflammation, as measured by CRP levels, is a mediator of physical activity's effect on depression at the within-subjects level in a population of older adults. We hypothesized that higher deviations in physical activity (i.e., deviations from the mean) at Wave 1 would predict lower inflammation deviations at Wave 2, which in turn, would be associated with lower depression symptoms deviations at Wave 3.

Our secondary aim was to estimate the stable, trait-like associations among physical activity, inflammation, and depression across the 3 waves of data collection (i.e., between-person component). We hypothesized that higher levels of physical activity across the measurement waves would be associated with lower CRP levels across measurement waves, and vice versa. Additionally, we hypothesized that higher levels of physical activity across the measurement

waves would report lower depression scores across measurement waves, and vice versa. Furthermore, we hypothesized that individuals with higher CRP levels across the measurement waves would report higher depression scores across measurement waves, and vice versa. Lastly, we aimed to explore other concurrent and lagged associations at the within-person level to gather a better understanding of the relations between physical activity, inflammation, and depression.

## Methods

### Participants

For this secondary analysis, data was used from the Health and Retirement Study (HRS). The HRS (Health and Retirement Study) is sponsored by the National Institute on Aging (grant number NIA U01AG009740) and is conducted by the University of Michigan. Participants in HRS are adults born in the contiguous United States between 1931 and 1947. HRS provides a developing overview of the physical and mental well-being, insurance protection, economic condition, familial support structures, employment situation, and retirement preparations of an aging population in America. Over the years, the initial sample has been updated to include newer birth cohorts and additional measures have been incorporated to investigate new research questions [74]. Extensive documentation is available through the HRS website describing the study's design and methodology [75].

A psychosocial survey that captured self-reported depression [76], and biomarker assessment that included CRP levels [77], were introduced to the study in 2006. Biomarker data was collected for half of the core sample each wave, meaning that longitudinal biomarker data is available every four years. To increase the sample size for our analysis, we subsequently combined data from two cohorts and treated data from 2006 and 2008 as Wave 1, data from 2010 and 2012 as Wave 2, and data from 2014 and 2016 as Wave 3. Participants with at least one biomarker data point available were incorporated into the analysis. In general, participants were older adults between the ages of 50 and 100, tended to identify as female, were predominately White and of non-Hispanic origin, and graduated high school or college. See Table 1 for more details.

### Measures

**Physical activity.** Participants reported their light, moderate, and vigorous-intensity physical activity over the prior year using a self-report Likert scale from 1 (never) to 5 (every day). Consistent with prior HRS exercise research [78,79], we developed weighted variables for each level of physical activity intensity, which were then aggregated to form a composite physical activity measure with total scores ranging from 0–54. The weighting for physical activity intensity was categorized as follows: for light physical activity, the scale was 0 (never), 1 (1–3 times a month), 3 (once a week), 6 (more than once a week), and 12 (daily). Moderate activity weights were set at 0, 1.5, 4.5, 9, and 18, while vigorous physical activity weights were 0, 2, 6, 12, and 24, respectively. As an example, a participant who reported that over the last year, they participated in light physical activity daily (12), moderate physical activity more than once a week (9), and vigorous physical activity 1–3 times a month (2) would have a composite sum score of 23 (12 + 9 + 2 = 23).

**High sensitivity C-reactive protein (hsCRP).** Specific informed consent was obtained for the process of blood collection within HRS.

Blood samples were collected through a sterile lancet prick on the participant's finger, previously sanitized with alcohol. Blood droplets were then squeezed out and applied onto circles on chemically prepared filter paper. The blood spot card was air dried for 10–15 minutes, then

**Table 1. Demographic statistics.**

| Variable | |
|---|---|
| N | 13,461 |
| **Age (yrs)** | |
| Age (mean) | 68.21 |
| Age (SD) | 9.83 |
| Age (min) | 50 |
| Age (max) | 100 |
| **Gender (%)** | |
| Female | 58.69 |
| Male | 41.31 |
| **Race (%)** | |
| White/caucasian | 82.08 |
| Black/African American | 12.96 |
| Other | 4.95 |
| Did not respond | .01 |
| **Ethnicity (%)** | |
| Not Hispanic | 90.59 |
| Hispanic | 9.40 |
| Did not respond | .01 |
| **Education (%)** | |
| Limited high school | 20.21 |
| GED | 4.56 |
| High school graduate | 31.25 |
| Some college | 22.20 |
| College and above | 21.76 |
| Did not respond | .02 |

stored in foil envelopes containing a desiccant, placed into mailing containers, and sent to laboratories for analysis. This method ensured the preservation of specimen values without needing temperature regulation [77]. Over the years of HRS data collection, the labs conducting the blood assays for the study have changed. The blood assays for the 2006 and 2008 samples (Wave 1) were shipped to The University of Vermont and assayed for hsCRP using a standard enzyme-linked immunosorbent assay (ELISA) [77]. The blood assays for the 2010, 2012, 2014, and 2016 samples (Waves 2 and 3) were shipped to The University of Washington and assayed for hsCRP using a standard ELISA [80–82]. For all waves, the hsCRP assay had a lower limit of detection of 0.04 mg/L, with a within-assay imprecision of 8.1% and a between-assay imprecision of 11.0%.

**Center for epidemiologic studies depression scale (CES-D; [83]).** Depression within HRS was measured using a modified 8-item version of the CES-D [84]. The CES-D has been extensively applied in research on depression in older adults, and has demonstrated strong psychometric properties for use with this population [85,86]. Participants responded with a 'yes' or 'no' to questions about whether they had experienced the following during a significant portion of the previous week: 1) I felt depressed; 2) I felt everything I did was an effort; 3) My sleep was restless; 4) I was happy; 5) I felt lonely; 6) I enjoyed life; 7) I felt sad; 8) I could not "get going". For each participant, a cumulative depressive symptom score was calculated by summing the "yes" answers to items 1, 2, 3, 5, 7, 8, and the "no" answers to items 4 and 6, resulting in a depression score from 0 to 8. Within the HRS study, the CES-D has shown strong internal consistency, ($\alpha$ = 0.80 to 0.83), for a brief measure. Individuals

reporting three or more depressive symptoms were categorized as experiencing significant depressive symptoms - a threshold identified to yield results analogous to the 16-symptom threshold of the well-validated 20-item CES-D scale [87].

## Statistical analysis

For a variety of reasons, we used RI-CLPMs to examine associations between physical activity, hsCRP levels, and CES-D scores over time. Within RI-CLPMs, the autoregressive paths quantify the stability of a particular variable over time by measuring the extent to which an individual's previous measurement on a variable (e.g., physical activity level at Wave 1) predicts their subsequent measurement on the same variable (e.g., physical activity level at Wave 2). This allowed us to assess the consistency of each variable in the model. A distinguishing feature of the RI-CLPM is that it also incorporates a random intercept for every latent variable, capturing stable individual differences and distinguishing changes that occur at the within-person level over time. This approach ensures that the cross-lagged paths in the model reflect these within-person changes, adjusted for between-person variance. This aspect of the RI-CLPM is subsequently useful in mechanisms of change research, such as the current study, because it allowed us to predict change within an individual and answer our relevant research questions (e.g., "Does a person who experiences lower than usual inflammation at Wave 1 experience higher than usual depression at Wave 2?") [72]. Moreover, a RI-CLPM allowed us to examine the direction of potential causality between our variables due to the model's inherent temporal sequencing. We chose to implement a RI-CLPM over the traditional Cross-Lagged Panel Model (CLPM) as recent critiques posit that CLPMs might amplify the magnitude of cross-lagged effects, which are critical for testing theoretical predictions [88].

Several methodological advantages further justify the use of RI-CLPM over other analytical approaches such as cross-lagged latent growth curve modeling (CL-LGCM) or random effects/multilevel regression models (MLM). First, RI-CLPM reduces bias from unmeasured stable confounders by incorporating a random intercept for each construct, effectively controlling for stable between-person characteristics (e.g., chronic health conditions, long-standing behaviors) [89]. This feature reduces the risk of confounding bias due to unmeasured time-invariant variables—an advantage not as explicitly addressed in standard cross-lagged models or typical multilevel regressions. Second, RI-CLPM is suitable for three-wave designs. While some complex growth models benefit from having four or more time points, RI-CLPM remains feasible and robust for three-wave designs [90]. This study's focus is on the within-person changes across these three points in time rather than on the shape of a growth trajectory over many waves. By leveraging three time points, we can still capture meaningful within-person dynamics and examine temporal ordering while partialing out stable individual differences. Third, although multilevel (random effects) regression is a valuable tool for hierarchical data, it does not inherently model the time-specific cross-lagged links among physical activity, inflammation, and depression. Additionally, MLM does not as cleanly distinguish within-person fluctuations from between-person differences, which is a core objective of this study [91]. Together these points underscore why RI-CLPM provides the most direct and theoretically aligned approach to examining the temporal relationships among physical activity, inflammation, and depression. Adopting CL-LGCM or standard MLM in this study could obscure the within-person reciprocal processes, potentially conflating them with between-person variability or long-term growth patterns.

We tested two RI-CLPMs in the current study using methods similar to prior work in our lab [92]: 1) a random intercept cross-lagged panel model with no constraints on the lagged relations, grand means, and covariance structures (RI-CLPM); and 2) a random intercept cross-lagged panel model with autoregressive relations constrained to be equal, grand means

constrained to be equal, covariance structures constrained to be equal, and cross-lagged relations constrained to be equal (RI-CLPM: constrained). We evaluated both models and ultimately chose the one with the optimal fit for deeper analysis. In the case of similar model fit, we selected the most parsimonious (i.e., fewest estimated paths) model. Imposing constraints on the models can aid in achieving model convergence and simplify interpretation, especially when there's no a priori expectation of varying effects across waves, as was the case in our study [88]. All effects are presented using standardized parameters. The models employed full information maximum likelihood estimation to handle missing data.

In line with previous research, raw hsCRP measurements were log-transformed [77,93] before analysis due to the highly skewed distribution of CRP values. Additionally, consistent with prior research, hsCRP values exceeding 10 mg/l, considered outliers and indicative of potential acute inflammation from infection or injury (N = 1,248; 5.77%), were removed [93]. All data analyses were conducted using R, version 4.3.3 [94].

### Ethics statement

The data used in this study are from the Health and Retirement Study (HRS), which is publicly available and conducted by the University of Michigan. The HRS data are de-identified, and all analyses were conducted in accordance with the terms and conditions for using the dataset. The data were originally accessed for research purposes on March 1st, 2023. This study was exempt from IRB review as it involves the analysis of publicly available, de-identified data. The authors did not have access to information that could identify individual participants during or after data collection. The original HRS study received ethical approval from the University of Michigan's Institutional Review Board, and informed written and verbal consent was obtained from all participants. The HRS biomarker data used in this study are considered sensitive health data and are released to researchers who qualify for access only through a supplemental registration system through the HRS website. Additional written consent was obtained from participants who provided blood samples for the HRS biomarker data.

## Results

### Descriptive statistics

Means, standard deviations, medians, minimums, maximums, and sample sizes for CRP levels, CES-D scores, and weighted physical activity sum scores across waves are shown in Table 2. Bivariate correlations are presented in Fig 1. On average, participants in all waves of the study reported CES-D scores below the clinical cut-off of ≥3 used in prior research to indicate possible depression [87,93]. Additionally, on average participants in all waves of the study exhibited medium CRP levels (1–3 mg/L) according to the American Heart Association and Centers for Disease Control and Prevention cut-off recommendations (low = <1 mg/L; high = > 3 mg/L) [95]. The average composite score for physical activity was similar across waves, with mean scores of 16.07, 14.71, and 14.46. This score could indicate a combination of activities at different intensities, but leaning more towards the lower end of the activity spectrum. For example, it might reflect engaging in light physical activity and moderate physical activity more than once a week and no vigorous activity. It could also represent engaging in light activity once a week, moderate activity once a week, and vigorous activity more than once a week.

### Model fit

Two different RI-CLPM models were evaluated for their goodness of fit, both with log-transformed hsCRP values (see Table 3). The baseline RI-CLPM provided a poor fit for the data, $\chi^2 = 5,715.346$, df = 3, p = 0.000, RMSEA = 0.306 and CFI = 0.794. The log-transformed

**Table 2. Descriptive statistics for physical activity, CRP levels, and CESD scores for each wave.**

| Variable | M | SD | median | min | max | n |
|---|---|---|---|---|---|---|
| Wave 1 Physical Activity | 16.07 | 10.77 | 15.00 | 0.00 | 54.00 | 13,442 |
| Wave 2 Physical Activity | 14.71 | 10.55 | 14.00 | 0.00 | 54.00 | 17,083 |
| Wave 3 Physical Activity | 14.46 | 10.64 | 13.50 | 0.00 | 54.00 | 15,900 |
| Wave 1 CRP | 2.61 | 2.28 | 1.80 | 0.02 | 10.00 | 10,863 |
| Wave 2 CRP | 2.40 | 2.15 | 1.67 | 0.05 | 10.00 | 13,312 |
| Wave 3 CRP | 2.26 | 2.21 | 1.52 | 0.02 | 10.00 | 12,764 |
| Wave 1 CES-D | 1.41 | 1.95 | 1.00 | 0.00 | 8.00 | 13,323 |
| Wave 2 CES-D | 1.52 | 2.03 | 1.00 | 0.00 | 8.00 | 16,741 |
| Wave 3 CES-D | 1.50 | 2.01 | 1.00 | 0.00 | 8.00 | 15,572 |

CRP = C-reactive protein; CES-D = Center for Epidemiologic Studies Depression Scale.

constrained RI-CLPM had a much better fit than the baseline RI-CLPM, $\chi2\Delta$ = 3925.6, df =23, p = 1. The constrained model had a lower $\chi2$, higher CFI, and lower RMSEA (see Table 3). It also returned lower AIC and BIC values compared to the constrained model. Given these considerations, the log-transformed constrained RI-CLPM was retained, $\chi2$ = 1789.782, df = 26, p = 0.000, RMSEA =0.058 and CFI =.936 (Fig 2).

## Model parameters for best fitting model

**Between-person results.** Significant relations were found amongst all the variables at the between-person level (Fig 3). The between-person association between physical activity and CRP levels was moderate and negative ($\beta$ = −0.40, $SE$ = 0.087, $p < 0.001$), indicating that individuals with higher levels of physical activity across the measurement waves had lower CRP levels across measurement waves, and vice versa. The between-person association between physical activity and CES-D scores was also moderate and negative ($\beta$ = −0.48, $SE$ = 0.137, $p < 0.001$), indicating that individuals with higher levels of physical activity across the measurement waves reported lower CES-D scores across measurement waves, and vice versa. The between-person association between CRP levels and CES-D scores was weaker and positive ($\beta$ = 0.19, $SE$ = 0.20, $p < 0.001$), indicating that individuals with higher CRP levels across the measurement waves reported higher CES-D scores across measurement waves, and vice versa.

**Within-person results.** The wave-to-wave autoregressive paths were moderate for CRP levels between Wave 1 and Wave 2 ($\beta$ = 0.30, $SE$ = 0.022, $p < 0.001$) and between Wave 2 and Wave 3 ($\beta$ = 0.34, $SE$ = 0.022, $p < 0.001$), suggesting moderate stability in CRP across time even after accounting for stable-between person variation. The wave-to-wave autoregressive paths were moderate for physical activity deviations between Wave 1 and Wave 2 ($\beta$ = 0.24, $SE$ = 0.016, $p < 0.001$) and between Wave 2 and Wave 3 ($\beta$ = 0.23, $SE$ = 0.016, $p < 0.001$), suggesting moderate stability in physical activity across time. The wave-to-wave autoregressive paths were relatively small (but still significant) for CES-D deviation scores between Wave 1 and Wave 2 ($\beta$ = 0.13, $SE$ = 0.017, $p < 0.001$) and between Wave 2 and Wave 3 ($\beta$ = 0.13, $SE$ = 0.017, $p < 0.001$), suggesting that depression was not highly stable after accounting for the between-person variation across the measurement waves. Note that the autoregressive paths were constrained to be equivalent for each variable in these models.

There were two significant but weak within-person associations within Wave 1 (Fig 2). A small negative correlation existed for the physical activity deviation score and CES-D deviation score at Wave 1 ($\beta$ = -0.08, $SE$ = 0.174, $p < 0.001$), indicating that within Wave 1

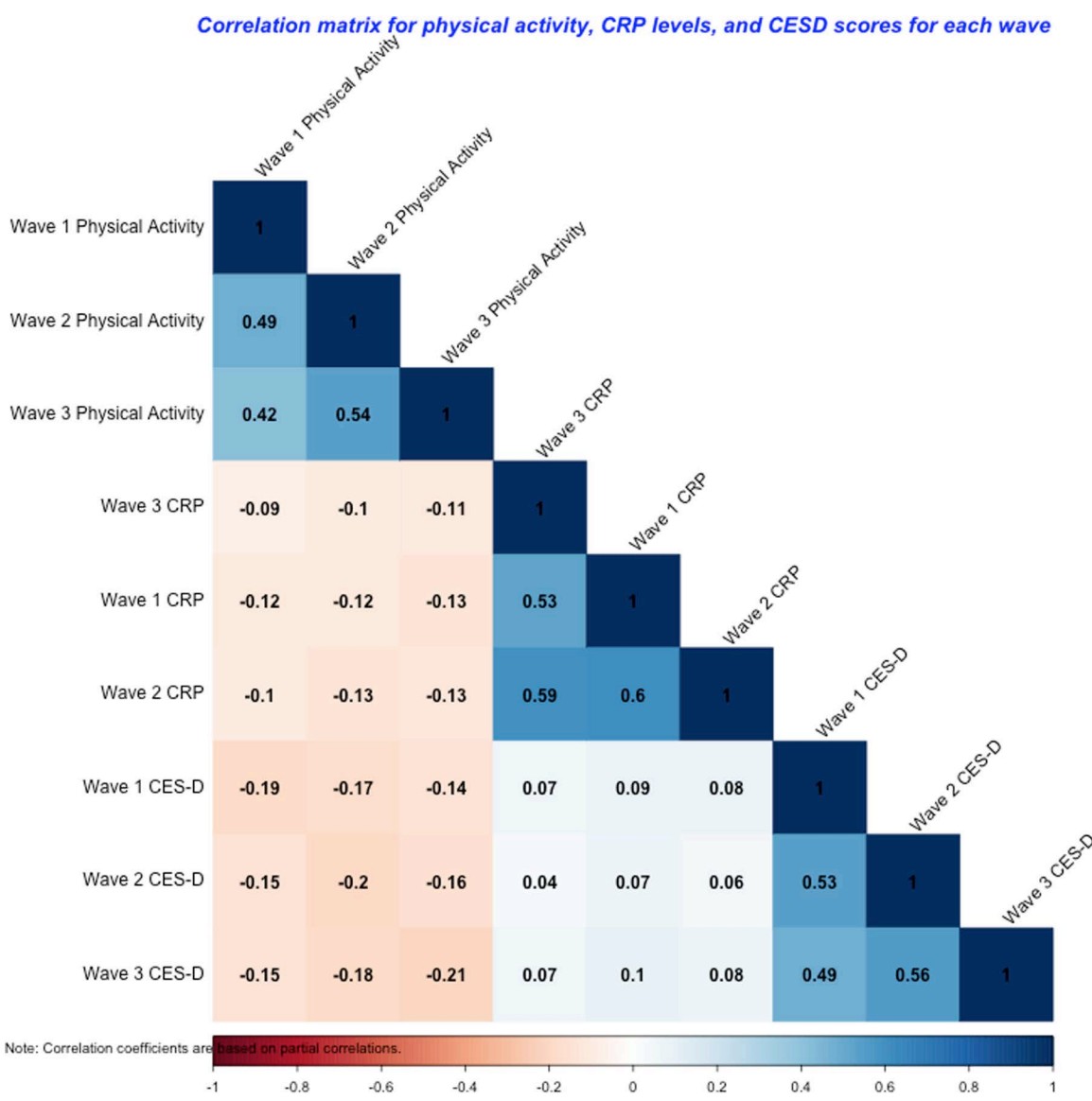

**Fig 1. Correlation matrix for physical activity, CRP levels, and CESD scores for each wave.** CRP = C-reactive protein; CES-D = Center for Epidemiologic Studies Depression Scale.

**Table 3. Model fit parameters for RI-CLPMs.**

| model | npar | chisq | df | p value | cfi | rmsea | aic | bic |
|---|---|---|---|---|---|---|---|---|
| RI-CLPM | 51 | 5715.346 | 3 | 0 | 0.794 | 0.306 | 630510.384 | 688681.139 |
| RI-CLPM: constrained | 51 | 1789.782 | 26 | 0 | 0.936 | 0.058 | 626538.820 | 626760.651 |

npar = number of estimated model parameters; cfi = comparative fit index; rmsea = root mean square error of approximation; aic = Akaike information criterion; bic = Bayesian information criterion; CRP values in both models were log transformed.

individuals who were more physically active than average tended to have a lower depression score than average, while those that were less active than average tended to have a higher depression score than average. Additionally, contrary to what prior research might suggest,

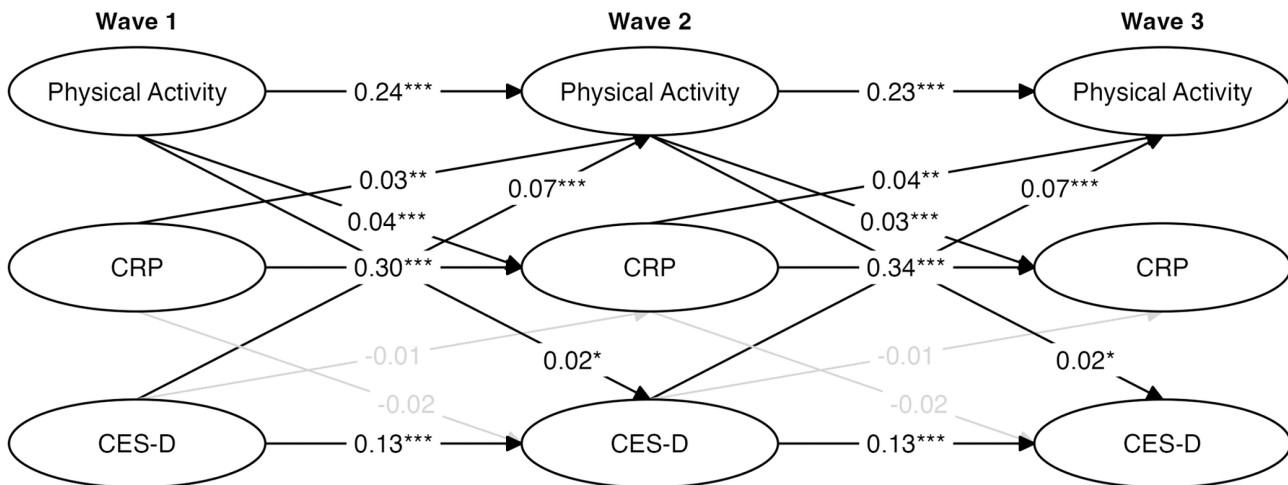

**Fig 2. Within-person standardized parameters for log-transformed constrained random intercept cross-lagged panel model (RI-CLPM).** * p < .05, ** p < .01, *** p < .001.

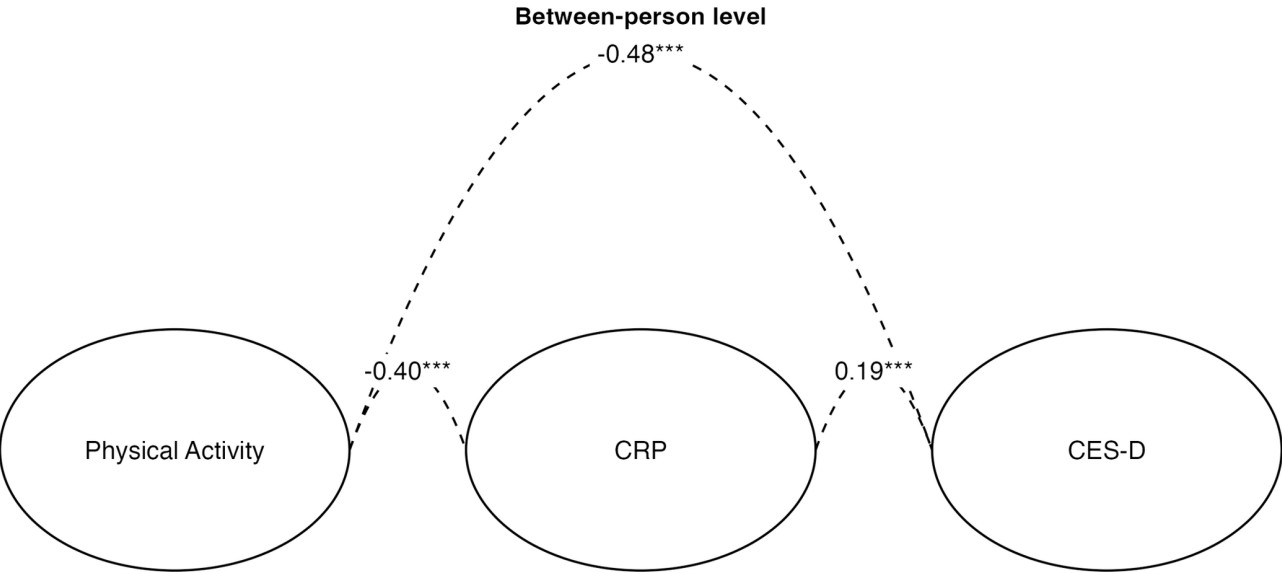

**Fig 3. Between-person correlations for log-transformed constrained Random Intercept Cross-Lagged Panel Model (RI-CLPM).** CRP = C-reactive protein; CES-D = Center for Epidemiologic Studies Depression Scale; *** p < .001.

there was a small positive within-person correlation between physical activity deviation score and CRP deviation levels at Wave 1 ($\beta$ = 0.06, *SE* = 0.109, p < 0.001), indicating that individuals who were more physically active than average tended to have higher CRP levels than average, and vice versa.

When examining cross-lagged relations across the assessment waves at the within-person level, no significant cross-lagged paths were observed between physical activity at Wave 1, CRP levels at Wave 2, and CES-D scores at Wave 3. In reference to the effect size benchmarks

provided by Orth et al. [96], several small (.03) and medium (.07) cross-lagged associations were found at the within-person level (Fig 2), however, they were all in the opposite direction from what we expected based on prior work. A small positive within-person lagged correlation existed between physical activity at Wave 1 and CES-D scores at Wave 2 (β = 0.02, *SE* = 0.002, p < 0.05), indicating that individuals with higher physical activity than usual at Wave 1 tended to have higher CES-D scores than usual at Wave 2.

A similar association was observed between Wave 2 and Wave 3 since the cross-lagged associations were constrained to be equal in the model (Fig 2). A small positive within-person lagged correlation also existed between physical activity at Wave 1 and CRP levels at Wave 2 (β = 0.04, *SE* = 0.001, p < 0.001), indicating that individuals with higher physical activity than usual at Wave 1 tended to have higher CRP levels than usual at Wave 2. A similar association was observed across both lags since the cross-lagged associations were constrained to be equal in the model (Fig 2). Finally, a moderate positive within-person lagged correlation also existed between CES-D scores at Wave 1 and physical activity at Wave 2 (β = 0.07, *SE* = 0.054, p < 0.001), indicating that individuals with higher CES-D scores than usual at Wave 1 tended to have higher physical activity than usual at Wave 2. A similar association was observed across both lags since the cross-lagged associations were constrained to be equal in the model (Fig 2).

**Supplementary analyses.** Two additional constrained RI-CLPM models using log-transformed hsCRP values were conducted to examine if there would be any differences in findings if looking at only moderate- and vigorous-intensity physical activity, or only vigorous physical activity. Both models fit the data well (see Table 4) and returned results similar to the model that included light-, moderate-, and vigorous-intensity physical activity (Figs 4 and 5).

## Discussion

The present study sought to investigate the longitudinal relations between physical activity, inflammation as measured by hsCRP levels, and depressive symptoms among over 13,000 older adults within the HRS. Our findings contribute to the growing body of evidence suggesting that physical activity has a multifaceted role in the modulation of inflammatory processes and the management of depressive symptoms. Our analysis, employing random intercept cross-lagged panel models (RI-CLPM), illuminated the dynamics between physical activity, hsCRP levels, and depressive symptoms over time. Contrary to our primary hypothesis, we did not observe a within-person mediation effect of inflammation on the relation between physical activity and depression. That is, when individuals' physical activity was lower than usual, they did not experience a subsequent increase in inflammation four years later. Similarly, when inflammation was higher than usual, this did not predict increases in depression at the next assessment.

Although we did not observe cross-lagged effects consistent with mediation at the within-person level, between-person analyses found robust associations among the stable (or trait)

**Table 4. Model fit parameters for moderate and vigorous, and vigorous physical activity only, RI-CLPMs.**

| model | npar | chisq | df | p value | cfi | rmsea | aic | bic |
|---|---|---|---|---|---|---|---|---|
| Moderate and Vigorous RI-CLPM: constrained | 51 | 1454.310 | 26 | 0 | 0.948 | 0.052 | 612176.185 | 612398.017 |
| Vigorous Only RI-CLPM: constrained | 51 | 1174.429 | 26 | 0 | 0.954 | 0.047 | 573801.501 | 574023.332 |

npar = number of estimated model parameters; cfi = comparative fit index; rmsea = root mean square error of approximation; aic = Akaike information criterion; bic = Bayesian information criterion; CRP values in both models were log transformed.

**Within-person level**

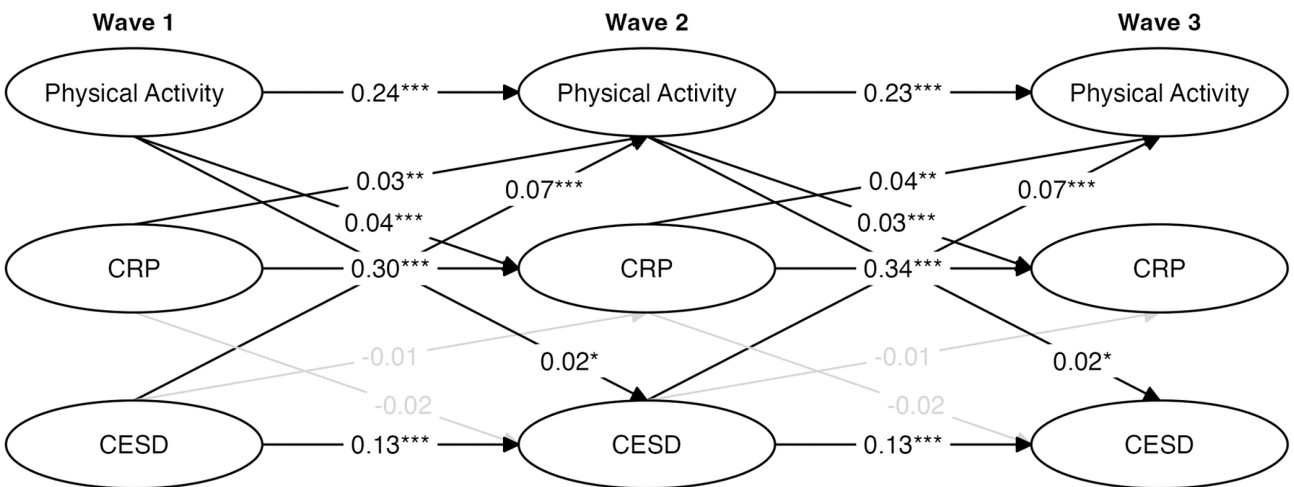

**Fig 4. Within-person standardized parameters for log-transformed constrained Random Intercept Cross-lagged Panel Model (RI-CLPM) of moderate and vigorous physical activity.** CRP = C-reactive protein; CES-D = Center for Epidemiologic Studies Depression Scale. * p < .05, ** p < .01, *** p < .001.

**Within-person level**

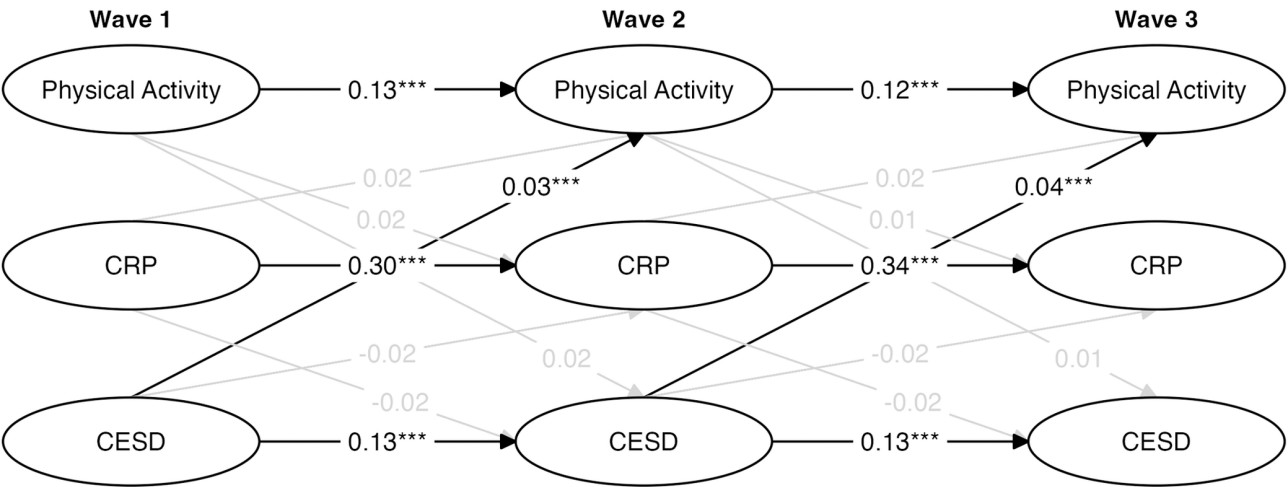

**Fig 5. Within-person standardized parameters for log-transformed constrained Random Intercept Cross-lagged Panel Model (RI-CLPM) of vigorous physical activity.** CRP = C-reactive protein; CES-D = Center for Epidemiologic Studies Depression Scale. * p < .05, ** p < .01, *** p < .001.

components of physical activity, inflammation, and depression. More specifically individuals with higher levels of physical activity across the measurement waves had lower CRP levels across measurement waves, and vice versa. Additionally, individuals with higher levels of physical activity across the measurement waves reported lower CES-D scores across measurement waves, and vice versa. Finally, individuals with higher CRP levels across the measurement waves reported higher CES-D scores across measurement waves, and vice versa. Overall,

these findings support the research which suggests that people who generally engage in higher levels of physical activity tend to have lower levels of inflammation and depressive symptoms compared to their less active counterparts.

We also found several other significant cross-lagged relations that were not aligned with the associations found in prior research between physical activity, inflammation, and depression. Contrary to what we hypothesized based on prior work - that there would be negative within-person correlations between physical activity and both CRP levels and CES-D scores across time points - the observed cross-lagged relations uniformly exhibited small but positive correlations. This suggests that an individual who reports greater physical activity than usual at an earlier Wave experienced increases in subsequent CRP levels and CES-D scores at later time points. This pattern held for all cross-lagged relations between physical activity and both CRP levels and CES-D scores, a pattern that is inconsistent with the predicted negative correlations.

These results may simply point to the possibility that CRP does not mediate the relation between physical activity and depression as recent research suggests may be the case with other inflammatory biomarkers [16]. It also is possible that our approach to testing this mediational pathway fell short. For example, while prior work looking at longitudinal associations between depression and inflammation have utilized time intervals as long as six years [97], it is possible that the time lag of four years between measurement waves is too large to capture pertinent changes in our three variables of interest. Within HRS, the self-report scale for physical activity is anchored over the past year, the CES-D is anchored over the past week, and hsCRP levels are captured via a blood draw that represents the participant's CRP levels at one moment in time. Research has consistently highlighted the heterogeneity of depression [98] and how substantial fluctuations in symptom networks within individuals can shift within as little as 90 days [99]. Indeed, intensive measurements of individuals with MDD in their daily lives reveal that symptoms fluctuate significantly, showing more variability within hours of a day than over weeks or months [100–103].

This dynamic nature of MDD symptoms could lead the current study to miss significant changes in depression symptoms due to physical activity or inflammation changes within a 4-year-time lag. Consistent with this possibility, we found that CES-D scores demonstrated the least amount of stability in our models (see the autoregressive paths in Figs 2, 4, and 5). It is possible that changes between physical activity, inflammation, and depression could have occurred at the within-person level, but sometime else during the 4-year time lag that simply wasn't captured by HRS measurements. Future studies would benefit from utilizing more frequent sampling techniques to better capture a potential mediational path between physical activity, inflammation, and depressive symptoms.

The absence of a significant mediating effect of hsCRP on the relation between physical activity and depressive symptoms aligns partially with the mixed outcomes of prior research. While some studies have identified clear anti-inflammatory effects of physical activity and linked these effects to improvements in depression outcomes [65,66], others have reported more nuanced relations. For instance, the findings by Hennings, [62], Krogh [64], and Rethorst [63] highlight the variability in physical activity's impact on inflammatory markers among depressed populations. Our findings suggest that while physical activity may still be beneficial for mental health, the underlying mechanisms may extend beyond simple modulation of inflammation levels. Our results could be further complicated by the use of older adults in our sample, where studies of inflammation and physical activity in the elderly have shown mixed results [58,59].

Another possibility for why the cross-lagged associations deviated from our expectations is recent evidence suggesting that only a subgroup of depressed patients exhibit a low-grade

inflammatory state (i.e., CRP > 3.0 mg/L) [43], which supports the hypothesis that while inflammation may play a role in the onset of certain forms of depression, it may not universally apply to all manifestations of the disorder [104]. A recent meta-analysis that included 37 studies with 13,541 depressed patients and 155,728 controls found that only about one-quarter of depressed patients showed evidence of low-grade inflammation and more than half of the participants exhibited mildly elevated CRP levels. This suggests the possibility that persistent, mild inflammation might indicate a unique subset of MDD characterized by its own cause, progression, and response to treatment [43,105,106]. It also may point to potential protective factors for depression, which may include lifestyle variables such as physical activity. Indeed, a recent meta-analysis investigating risk and protective factors of depression in adults over the age of 65 found that engaging in more physical activity was protective against depression [107]. It is subsequently possible that our current model was ill-suited to capture this possible sub-group of depressed patients' longitudinal relation with inflammation and depression if the brevity of the study's CES-D measure translated to poor construct validity with regard to measuring this specific sub-group of depressed patients.

Alternatively, it is possible that the physical activity self-report measure within HRS was too broad and lacked the specificity to detect how different physical activity subgroups differentially impact different inflammatory biomarkers such CRP. Indeed, a review by Eyre and Baune [60] found that positive anti-inflammatory clinical effects of physical activity for unipolar depression may vary depending on the physical activity subtype (e.g., aerobic, resistance, flexibility, mind-body). For example, a study comparing 10 weeks of resistance and aerobic physical activity with control in 103 adults found that CRP levels were reduced more by resistance training compared to aerobic training. Within HRS, only the intensity of physical activity was reported and not the specific type of physical activity. It is therefore possible that our model was ill-equipped to capture the intricacies of the different effects various subtypes of physical activity may have on inflammation and/or depression.

Another possibility for why our within-person results were not consistent with our theoretical framework is that perhaps the composite sum scores created to measure physical activity did not accurately capture the differences between different levels of physical activity intensity. Research has frequently pointed to differential impacts of physical activity on depression [5,6,14,15] depending on the intensity of activity. To account for the potential that the light and/or moderate-intensity physical activity was being over-represented in our weighted composite sum score, and to see if the intensity of physical activity on its own made an impact on the results, we ran supplementary analyses where physical activity was measured as the sum score of moderate and vigorous-intensity physical activity (Fig 4) and only vigorous-intensity physical activity (Fig 5). However, the results were nearly identical to the original model utilizing a weighted composite sum score of all three physical activity intensities (Fig 2), suggesting that perhaps the weighting methodology is not the driving force behind the null results.

It was also notable that the model continued to fit well even after constraining the autoregressive path for each variable and cross-lagged relationships between variables to be equal over time. Constraints on those associations did not hurt overall model fit, suggesting there were not strong differences in the strength of the cross-lagged associations over time. Prior work examining the linkages between physical activity and CRP did not disentangle the between- and within-person variance, so the stable trait-like variance could have been driving previously observed associations. Indeed, in the current study, we found strong associations for activity level and CRP at the between-subjects level but not at the within-subjects level.

Lastly, while hsCRP serves as an indicator of systemic inflammation, it's important to recognize that it is only one among many markers, and likely unable to fully encapsulate the complexity of inflammatory processes on its own. As such, it is possible that while strong evidence

links increased physical activity to reductions in inflammation [49–51,53,54], our findings add to the research that shows inconsistent associations between physical activity and CRP specifically [52,59]. Additionally, while our study focuses on CRP as a marker of chronic inflammation, we acknowledge that elderly adults often exhibit low-grade chronic inflammation involving multiple pathways and biomarkers beyond CRP [108]. This "inflammaging" is characterized by the dysregulation of both pro-inflammatory and anti-inflammatory mechanisms, which may influence how chronic inflammation responds to exercise and other interventions over time [68,109,110]. It is possible that other biomarkers, such as IL-6 and TNF-α, may better capture these chronic inflammatory mechanisms and their long-term changes in response to physical activity or other behavioral health interventions [68,108,110]. While ideally we would have measured multiple biomarkers in the current study, only CRP was collected within the HRS in at least three waves of data collection, which is needed for a RI-CLPM [73]. IL-6 and TNF-α were only introduced within HRS in 2014 and 2016, respectively, whereas CRP data spans from 2006–2016. Future research should concurrently measure multiple inflammatory biomarkers to better understand how exercise and chronic inflammation influence depression symptoms in elderly populations, identifying which biomarkers might mediate the positive effects of physical activity on depression.

While the results failed to support our hypotheses at the within-person level, we did find strong evidence in line with prior research at the between-person level. Our analysis revealed significant and robust connections across all examined variables at the individual level. Specifically, a moderate negative correlation was observed between physical activity and CRP levels, suggesting that more active individuals tend to have lower CRP levels. Similarly, physical activity was negatively associated with CES-D scores, indicating that higher activity levels correlate with lower depression scores. Conversely, a weaker positive link was found between CRP levels and CES-D scores, implying that higher CRP levels may be associated with higher depression scores. Furthermore, each of these associations were found using a longitudinal dataset with over 13,000 subjects. These findings add to the current literature and offer strong support that these constructs are linked at the between-person level. That is, the trait levels of these variables appear to be robustly associated with each other, consistent with prior research in this area.

While this study had several strengths, including longitudinal assessment with three-time points and a large population-based sample, there were also several limitations. As mentioned previously, perhaps the biggest limitation was that the time lags for the RI-CLPM were four years apart, making it difficult to detect the often short-term changes in depression symptoms, inflammation, and the impact of physical activity on both of these variables. Additionally, the physical activity and depression measures were both brief self-report measures. Due to the vast heterogeneity of depression and a broad range of subtypes of physical activity, more objective, frequent, and comprehensive measures would greatly improve the ability to detect effects between the variables of interest. The physical activity measure within HRS was also lacking information on the subtypes of physical activity participants were engaging in, which research has shown has distinct effects on depression and inflammation. Furthermore, hsCRP is one of many different potential biomarkers for inflammation. It offers a glimpse into systemic inflammation, but the intricate nature of inflammation cannot be fully captured by a single metric alone and future studies would benefit from examining additional inflammatory markers. Additionally, within our model, other confounding lifestyle factors that might influence both inflammation and depression were not fully accounted for, such as diet, sleep quality, and comorbid health conditions. Lastly, our study findings are based on an older adult population, which may limit generalizability to younger demographics or individuals with different health statuses.

In conclusion, the current study elucidates the links between physical activity, inflammation, as indicated by hsCRP levels, and depressive symptoms among older adults, revealing differential associations at the between-person (trait) and within-person levels. The results of our constrained RI-CLPM fit the data well and revealed significant between-person relations among physical activity, CRP levels, and depressive symptoms. Higher physical activity was associated with lower CRP levels and depressive symptoms, suggesting a protective role of physical activity against inflammation and depressive symptoms. Additionally, individuals who reported higher levels of physical activity across time reported lower CES-D scores across time, suggesting that those who exercised more were less depressed. And lastly, individuals with higher CRP levels reported more depressive symptoms, suggesting that higher levels of inflammation is associated with higher depressive scores. Unfortunately, we failed to find support for inflammation, as measured by hsCRP, being a mediator of physical activity's impact on depression and many of the significant cross-lagged relations produced by our model were unexpected given the findings of prior work. Nonetheless, the significant findings at the between-person level with a large longitudinal sample of older adults provide evidence that future longitudinal research investigating the relations between physical activity, inflammation, and depression is needed to fully understand the causal relations among these variables and how these relations may or may not unfold over time.

## Author contributions

**Conceptualization:** Andrew Levihn-Coon, Christopher G. Beevers.

**Data curation:** Andrew Levihn-Coon, Christopher G. Beevers.

**Formal analysis:** Andrew Levihn-Coon, Christopher G. Beevers.

**Investigation:** Andrew Levihn-Coon, Christopher G. Beevers.

**Methodology:** Andrew Levihn-Coon, Christopher G. Beevers.

**Project administration:** Andrew Levihn-Coon, Christopher G. Beevers.

**Resources:** Andrew Levihn-Coon.

**Software:** Andrew Levihn-Coon, Christopher G. Beevers.

**Supervision:** Christopher G. Beevers.

**Validation:** Andrew Levihn-Coon, Christopher G. Beevers.

**Visualization:** Andrew Levihn-Coon, Christopher G. Beevers.

**Writing – original draft:** Andrew Levihn-Coon.

**Writing – review & editing:** Andrew Levihn-Coon, Jasper A.J. Smits, Christopher G. Beevers.

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
