## [Decision Letter · Decision Letter 0]

2 Oct 2024

PMEN-D-24-00384

The longitudinal relation between physical activity, inflammation, and depression: A mediational analysis using the Health and Retirement Study

PLOS Mental Health

Dear Dr. Levihn-Coon,

Thank you for submitting your manuscript to PLOS Mental Health. After careful consideration, we feel that it has merit but does not fully meet PLOS Mental Health’s publication criteria as it currently stands. Therefore, we invite you to submit a revised version of the manuscript that addresses the points raised during the review process.

ed on PLOS Mental Health’s publication criteria  and not, for example, on novelty or perceived impact.

We look forward to receiving your revised manuscript.

Kind regards,

Gareth Hagger-Johnson

Academic Editor

PLOS Mental Health

Journal Requirements:

1. We noticed you have some minor occurrence of overlapping text with the following previous publication(s), which needs to be addressed:

- https://doi.org/10.3389/fpsyg.2020.571943

- doi:10.1016/j.psyneuen.2018.05.035.

In your revision ensure you cite all your sources (including your own works), and quote or rephrase any duplicated text outside the methods section. Further consideration is dependent on these concerns being addressed.

Additional Editor Comments (if provided):

The manuscript is potentially suitable for publication in PLOS Mental Health, if you can address concerns raised by the reviewers.

Reviewers' comments:

Reviewer's Responses to Questions

**Comments to the Author**

1. Does this manuscript meet PLOS Mental Health’s publication criteria ? Is the manuscript technically sound, and do the data support the conclusions? The manuscript must describe methodologically and ethically rigorous research with conclusions that are appropriately drawn based on the data presented.

Reviewer #1: Yes

Reviewer #2: Yes

Reviewer #3: Yes

2. Has the statistical analysis been performed appropriately and rigorously?

Reviewer #1: Yes

Reviewer #2: No

Reviewer #3: Yes

3. Have the authors made all data underlying the findings in their manuscript fully available (please refer to the Data Availability Statement at the start of the manuscript PDF file)?

Reviewer #1: Yes

Reviewer #2: Yes

Reviewer #3: Yes

4. Is the manuscript presented in an intelligible fashion and written in standard English?

Reviewer #1: Yes

Reviewer #2: Yes

Reviewer #3: Yes

5. Review Comments to the Author

Reviewer #1: The manuscript entitled "The longitudinal relation between physical activity, inflammation, and depression: A

mediational analysis using the Health and Retirement Study" has a very interesting topic. It is well written.

However, references should be up date with more recent evidence and also its related discussion section.

Reviewer #2: PMEN-D-24-00384 notes

This study aimed to examine the between- and within-persons associations among physical activity, inflammation, and depression symptoms among older adults. The strengths of this study include the use of a large dataset with three waves of repeated measures and examination of both between- and within-persons associations.

1. The Introduction may benefit from more precise identifications of literature gaps that the current investigation addresses. For example, the review of the effects of physical activity on inflammation (the paragraph starting on line 123) led to the conclusion that “more research is needed to investigate if physical activity reduces inflammation in depressed individuals”, leaving readers to expect the current investigation would target individuals with depression, but this isn’t the case. In a similar vein, the theoretical motivation for examining the associations among physical activity, inflammation, and depression in older adults is lacking. It is unclear why examining the current questions among older adults is important and why similar hypotheses, as expected by the reviewed evidence from younger populations, are expected in the current sample of older adults.

2. As the novel piece of the current investigation is between- vs. within-persons associations among physical activity, inflammation, and depression, the Introduction can benefit from theoretical reasons for this distinction (rather than simply that use of RI-CLPM allows for it). That is, what are we gaining from examining within-persons associations that we are not already gaining from the less resource-intensive investigations of between-persons associations as well as RCT that allows establishment of causality?

3. There are three major methodological concerns:

(a) Given the 3 timepoints of data, there are more fitting statistical methods for modeling within-individual changes. Currently, RI-CLPM models T1 � T2 and then T2 � T3, AND these links are constrained to be the same, so the 3 timepoints of data is not being utilized to adequately. Yet, both MLM and DSEM are statistical models that can estimate trajectories of depression/inflammation across all 3 timepoints. Particularly, DSEM will allow you to estimate autoregressive links across all three timepoints for all variables, lagged-associations, as well as the associations among within-individual trajectories of inflammation (time � inflammation), trajectories of physical activity (time � activity), and trajectories of depression (time � depression), as well as how these activity trajectories may covary with depression trajectories mediated by inflammation trajectories. In my opinion, the use of RI-CLPM is not examining the kind of within-persons changes expected from data that has 3 datapoints (rather than 2).

(b) The way RI-CLPM is implemented raises some concerns. While important, model fits do not solely determine whether the model is appropriate for the research questions at hand. By nature of constraining the many estimates (grand mean, autoregressive, cross-lagged, and covariance structure all were constrained) model fit will increase merely because less is being asked from the model to estimate. But constraining those paths do not necessary make sense theoretically—e.g., grand means being constrained to equal requires the theoretical assumption that there’s no systematic change in the average level of inflammation, depression, and physical activity, but we know CRP increases with age in the U.S., depression increases in older adults, physical activity decreases in older adults; another example: constraining cross-lagged links to equal requires the theoretical assumption that, for example, the prediction of physical activity on CRP is static over time, despite expected aging-related physiological declines. Given all these constraints imposed, it’s unclear how to draw conclusions from the findings, yet the Discussion does not interpret the findings in light of the theoretical implication of these constraints.

(c) There are somewhat inaccurate statements in describing within-persons associations in the context of RI-CLPM. As one example, in RI-CLPM, within-person associations refer to the cross-lagged associations, as you cannot really have within-person associations within a single wave, so statements like these “There were two significant but weak within-person associations within Wave 1 (Fig 2). A small negative within-person correlation existed between reported physical activity and reported CES-D scores at Wave 1 (β = -0.08, SE = 0.174, p < 0.001), indicating that within Wave 1 individuals who were more physically active tended to have a lower depression score, while those that were less active tended to have a higher depression score” (Line 359). The nature of these associations are more between-person, than within-person, associations, even if scores were person-mean centered.

4. HRS has other inflammatory cytokines (e.g., IL-6, TNF-a). Since the Introduction mentioned discrepant findings based on the marker assessed, would the authors consider parallel analyses with these cytokines?

5. The Discussion section largely explains the current null or opposite-than-hypothesized findings methodologically (e.g., 4 year lag might be too much to capture the more transient changes, sum score of physical activity obscuring findings). As described above, I agree there are methodological concerns, so along with the authors’ methodological interpretations of the findings, it does leave readers wondering whether the current methods are well-positioned to test the research questions authors set out to test from the get go. That said, I really think the examination of between- vs. within-individual associations is important, so would the authors consider using more fitting methods so that we have a better sense whether the unexpected/null results are due to suboptimal methods or in fact reflect the true nature of these relationships?

6. Minor: Line 134 suggests that “physical activity led to decrease in both pro-inflammatory markers and depressive symptoms” but that CRP was not available in that study, please state the pro-inflammatory markers assessed.

Reviewer #3: The authors investigated the relationship between physical exercise and CRP and depression in elderly adults.

They found a result in a cross-sectional study indicating that physical exercise reduces acute inflammation and depression scores.

However, in a longitudinal study, they found no association between exercise, CRP, and depression.

The authors discuss the limitations of the longitudinal study's long interval between assessment phases and possible differences between exercise protocols.

It is also important to highlight that elderly adults may have low-grade chronic inflammation that involves other markers and chronic inflammatory mechanisms. These mechanisms may be more noticeable in chronic inflammation and, therefore, more related to the influence of exercise over time.

The authors discussed and pointed out the strengths and weaknesses and made some suggestions.

I only suggest that the authors highlight that the individuals were elderly and, therefore, other markers could be more appropriate for assessing chronic inflammation, in addition to the lag between one phase and another throughout the study.

6. PLOS authors have the option to publish the peer review history of their article (what does this mean? ). If published, this will include your full peer review and any attached files.

**Do you want your identity to be public for this peer review?** For information about this choice, including consent withdrawal, please see our Privacy Policy .

Reviewer #1: **Yes: ** Elsa Vitale

Reviewer #2: No

Reviewer #3: No

---

## [Editor Report · Decision Letter 1]

22 Dec 2024

PMEN-D-24-00384R1

The longitudinal relations between physical activity, inflammation, and depression in the Health and Retirement Study

PLOS Mental Health

Dear Dr. Levihn-Coon,

Thank you for submitting your manuscript to PLOS Mental Health. After careful consideration, we feel that it has merit but does not fully meet PLOS Mental Health’s publication criteria as it currently stands. Therefore, we invite you to submit a revised version of the manuscript that addresses the points raised during the review process.

We look forward to receiving your revised manuscript.

Kind regards,

Gareth Hagger-Johnson

Academic Editor

PLOS Mental Health

Journal Requirements:

Additional Editor Comments (if provided):

This manuscript is nearly there.

Please address concerns raised by Reviewer 2 and/or be more explicit about alternatives for three-wave data (for example, cross-lagged latent growth curve modelling or random effect / multilevel regression are how I would approach this sort of data). When you have three or more waves, you can infer growth processes (linear or otherwise) rather than having separate regressions between every timepoint (see e.g. Bollen & Curran 2006 https://onlinelibrary.wiley.com/doi/book/10.1002/0471746096).

The autoregression approach is not popular with three or more waves available, so if you want to commit to using it here, I think you need to strengthen your argument and more fully acknowledge limitations + alternatives that you might consider in future research or that readers might consider with longitudinal data (aside from the cluttered diagrams it produces which is unfortunate). Can you do anything to improve/simplify the Figure and help readers visualise what message the Figure conveys? Relatively few readers will know about autoregressive models like this.

Please also address other relatively minor comments/suggestions.

Please upload a STROBE checklist with page numbers to a revised manuscript.

Many thanks,

Dr. Gareth Hagger-Johnson
---

## [Editor Report · Decision Letter 2]

20 Feb 2025

The longitudinal relations between physical activity, inflammation, and depression in the Health and Retirement Study

PMEN-D-24-00384R2

Dear Mr. Levihn-Coon,

Thank you for a good quality revision which addresses the reviewer's comments, and responds to my suggestions. It is well-written and convincing. On latent growth modelling, I just wanted to add that latent growth curve modelling in the generalised latent variable modelling framework can handle reciprocal effects, short term effects and within-person time-varying effects, any time period, stable within-person characteristics, and three wave designs. But I appreciate the additional text for readers, and it is clear from your response that you can defend your choice sufficiently well. There are always several ways to approach longitudinal data, each with advantages and limitations.

We are pleased to inform you that your manuscript 'The longitudinal relations between physical activity, inflammation, and depression in the Health and Retirement Study' has been provisionally accepted for publication in PLOS Mental Health.

Best regards,

Gareth Hagger-Johnson

Academic Editor

PLOS Mental Health
